# Changes in Functional Outcome and Quality of Life in Soft Tissue Sarcoma Patients within the First Year after Surgery: A Prospective Observational Study

**DOI:** 10.3390/cancers12020463

**Published:** 2020-02-17

**Authors:** Casper Saebye, Ali Amidi, Johnny Keller, Henning Andersen, Thomas Baad-Hansen

**Affiliations:** 1Department of Orthopedic Surgery, Sarcoma Centre of Aarhus University Hospital, Aarhus DK-8200, Denmark; 2Department of Experimental Clinical Oncology, Sarcoma Centre of Aarhus University Hospital, Aarhus DK-8200, Denmark; 3Unit for Psychooncology & Health Psychology, Department of Psychology, Aarhus University, Aarhus DK-8000, Denmark; 4Department of Neurology, Aarhus University Hospital, Aarhus DK-8200, Denmark

**Keywords:** soft tissue sarcoma, limb-sparing surgery, outcome, function, distress

## Abstract

Soft tissue sarcoma (STS) patients undergoing surgery may experience reduced functional outcome (FO) and quality of life (QoL) compared to the general population. The aim of this study was to evaluate the development of FO and QoL in patients with STS in the extremities within the first year after first-time limb-sparing surgery. Twenty-nine out of 40 eligible patients were included in the present study. QoL and FO were evaluated by questionnaires while FO was also evaluated by objective tests. Patients were assessed before surgery and at fixed time points within the first year after surgery. Patients with STS in the extremities had an average strength of 82.34% (95% CI: 68.57–96.11) of the expected strength at one year post surgery. Multivariate, repeated ANOVA showed statistically significant reductions in strength in the disease-affected extremity when compared with the healthy side after surgery. Multivariate, repeated ANOVA showed a statistically significant improvement in FO and QoL within the first year post surgery. Limb-sparing surgery for STS significantly reduced strength in the disease-affected extremity and generally reduced FO and QoL in the first months after surgery. Improvements were observed for FO and QoL at one year after surgery.

## 1. Introduction

Postoperative functional outcome and quality of life in soft tissue sarcoma patients have been of interest in sarcoma research for decades [1]. The introduction of limb-sparing surgery as the mainstay treatment has been shown to improve functional outcome when compared to amputation [2,3,4]. Studies indicate that soft tissue sarcoma patients who have undergone limb-sparing surgery have a reduced functional outcome compared to the general population [4]. A common method for measuring functional outcome in sarcoma patients has been the use of self-reported measures [4], such as the Musculoskeletal Tumor Society Score (MSTS) and the Toronto Extremity Salvage Score (TESS) [5,6]. Objective measurements of muscle strength and gait performance, in contrast, are only rarely performed. However, when objective measures have been undertaken, correlations between subjective measures, such as the MSTS and the TESS, and objective measurements tend to be low to moderate [7,8,9]. Isokinetic dynamometry (Biodex System 3 PRO^®^) enables evaluation of the muscle strength of all major muscle groups at the upper and lower extremities. Published normative data from the healthy population is available [10], enabling comparison of the muscle strength in sarcoma patients with appropriate healthy controls.

Previous studies have shown that reduced functional outcome has a negative impact on quality of life in soft tissue sarcoma patients [11,12,13]. In soft tissue sarcoma patients, quality of life has often been measured with health-related quality-of-life questionnaires such as the EORTC core quality of life questionnaire (QLQ-C30) or Short Form 36 (SF-36) [4,11,13]. A review by Tang et al. [4], however, suggested that psychological distress is an important aspect of quality of life, which is often overlooked in sarcoma research. In a recent study, Siracuse et al. [14] found an increased incidence of suicide in sarcoma patients compared with that in the general population. This highlights the importance of also investigating distress in this population.

The present study aims to investigate changes in objectively measured functional outcome for extremity soft tissue sarcoma patients within the first year after surgery and to investigate the association between functional outcome assessed by the TESS and the MSTS, and objective measurements of functional outcome, including isokinetic dynamometry. Furthermore, this study aims to clarify whether extremity soft tissue sarcoma patients evidence high levels of distress within the first year after surgery.

## 2. Results

A total of 29 out of 40 eligible patients were included in the study (Table 1). Eight patients declined to participate and in three patients the general condition was too poor to allow participation. No significant differences in the distribution of age or gender was found between included and excluded patients.

Frequency of patients who completed the objective measurements and questionnaires at each point in time are presented in Table 2. One patient was first included and measured at the one-month follow-up. A total of 14 patients did not complete objective measurements at one-month post-surgery. At three months, two patients had experienced recurrence and were excluded from that point onward, and eight patients did not wish to participate in the objective measurements. One year after surgery, two more patients had experienced recurrence of disease and were excluded, and one patient had a cerebral hemorrhage and was consequently excluded. Eleven patients did not wish to participate in the last measurement, but three agreed to complete the questionnaires.

### 2.1. Functional Outcome Measures

The results from the functional outcome measures (TESS, MSTS, isokinetic dynamometer, the 10-m walk test (10-MWT), the 9-hole peg test (9-HPT), and the 6-min walking test (6-MWT)) are presented in Table 3, Table 4 and Table 5. 

There was a statistically significant change over time in the TESS score, F(4, 80) = 9.02, *p* < 0.01 and the MSTS score, F(2, 26) = 11.85, *p* < 0.01. 

Patients evidenced a significant change in gait speed (10-MWT) over time, F(3, 29) = 3.13, *p* = 0.04. 

There was no significant difference between the disease-affected and the healthy side in 9-HPT, F(1, 19) = 0.13, *p* = 0.73, nor any statistically significant change over time, F(3, 9.37) = 0.10, *p* = 0.96.

A statistically significant difference was observed in muscle strength between the healthy and disease-affected side measured by the isokinetic dynamometer, F(1, 74) = 16.41, *p* < 0.01. Finally, there was a significant change over time in the average strength, F(3, 43) = 3.51, *p* = 0.02, but no statistically significant change in the exercise capacity of participants, F(3, 40) = 2.37, *p* = 0.09.

### 2.2. Distress Measures

Quality-of-life, insomnia severity, fatigue, and psychological distress measures (The World Health Organization-Five Well-Being Index (WHO-5), Insomnia Severity Index (ISI), Multidimensional Fatigue Inventory (MFI-20), Hospital Anxiety and Depression Score (HADS)) are presented in Table 3. A significant change over time in the HADS-Anxiety (F(4, 80) = 6.27, *p* < 0.01), MFI-20 Physical fatigue (F(4, 80) = 6.82, *p* < 0.01), and MFI-20 Reduced activity (F(4, 80) = 3.55, *p* = 0.01) was shown. However, no significant change over time was found in the WHO-5 (F(4, 80) = 1.96, *p* = 0.11), HADS-Depression (F(4, 80) = 1.30, *p* = 0.28), ISI (F(4, 80) = 1.17, *p* = 0.33), MFI-20 General fatigue (F(4, 80) = 0.45, *p* = 0.77), MFI-20 Mental fatigue (F(4, 80) = 0.93, *p* = 0.45), and MFI-20 Reduced motivation (F(4, 80) = 1.08, *p* = 0.37).

### 2.3. Associations between Self-Reported Outcomes and Objective Measurements

Associations between objective measurements (isokinetic dynamometer, 10-MWT, 9-HPT, and 6-MWT) and self-reported measures (TESS, MSTS, WHO-5, HADS, ISI, and MFI-20) are presented in Table 6.

Significant associations were found between the isokinetic dynamometer and the following self-reported outcomes: (1) TESS, (2) MFI-20 General fatigue, (3) MFI-20 Physical fatigue, (4) MFI-20 Reduced activity, and (5) MFI-20 Reduced motivation. Moreover, trends toward statistically significant associations were observed between the isokinetic dynamometer and the following: (1) WHO-5 and (2) HADS-Depression (Table 6).

Significant associations were also found between the 10-MWT and the following self-reported outcomes: (1) TESS, (2) MSTS, (3) MFI-20 General fatigue, (4) MFI-20 Physical fatigue, and (5) MFI-20 Reduced activity (Table 6).

In addition, significant associations were found between the 9-HPT and the following self-reported outcomes: (1) MSTS and (2) HADS-Anxiety. Furthermore, trends toward statistically significant associations were observed between 9-HPT and the following: (1) ISI and (2) MFI-20 General fatigue (Table 6).

Lastly, significant associations were found between the 6-MWT and the following self-reported outcomes: (1) HADS-Depression, (2) MFI-20 General fatigue, (3) MFI-20 Physical fatigue, and (4) MFI-20 Reduced motivation. Additionally, a trend toward a statistically significant association was observed between 6-MWT and MFI-20 Reduced activity (Table 6).

## 3. Discussion

In the present study, soft tissue sarcoma patients undergoing limb-sparing surgery experienced an initial decrease in functional outcomes, albeit improvements were observed within the first year following surgery. The disease-affected extremity muscle strength was lower compared to the healthy side one year after surgery. Furthermore, soft tissue sarcoma patients who underwent limb-sparing surgery seemed to overcome distress, such as reduced quality of life, insomnia, fatigue, and psychological distress within the first year, although increased levels of physical fatigue and reduced activity were also observed.

Patients experienced an initial postoperative decrease in functional outcome measured by the TESS and the MSTS. However, at one year after surgery, patients had regained their function. These findings are similar to previously reported results [15,16,17,18], and the tendency was also shown measuring functional outcome with 10-MWT in lower-extremity sarcoma patients. The 10-MWT has, to the best of our knowledge, not been used in sarcoma patients before. However, using other gait evaluations, similar results have been reported [19,20]. 

There was no statistically significant difference in dexterity of the disease-affected and the healthy side in upper-extremity sarcoma patients, as measured by the 9-HPT, nor statistically significant change over time. It is, however, important to factor in possible differences between the dominant and non-dominant side, which was not possible to do due to the small sample size of this study.

No studies have previously measured muscle strength with an isokinetic dynamometer in soft tissue sarcoma patients and compared the results with normative data. Soft tissue sarcoma patients’ muscle strength changed significantly over time, and patients had significantly reduced strength compared to the general population. Interestingly, it was found that soft tissue sarcoma patients also had significantly reduced strength before surgery when compared to the general population. In addition, a significant difference between the disease-affected and the healthy side was shown throughout the first year after surgery. This is in line with previous results by Tanaka et al. [8,9] who postoperatively compared the strength of thigh muscles at the disease-affected and the healthy side.

In the present study, the 6-MWT was regarded as an expression of the patients’ exercise capacity. No statistically significant change was observed on the 6-MWT; however, the results also showed a reduced exercise capacity prior to surgery, indicating that soft tissue sarcoma patients possibly have a reduced exercise capacity compared to the general population. With that in mind and the findings of reduced strength before surgery in soft tissue sarcoma patients, it is important, when evaluating functional outcome in soft tissue sarcoma patients, to take into account the fact that sarcoma patients can rarely be physically optimized for surgical treatment, since time spent waiting on initial surgery is a factor. Therefore, it is advisable to implement further research in preoperative rehabilitation of sarcoma patients, since previous studies for other cancer populations have shown significant positive effects of pre-operative training [21].

The MSTS and the TESS are commonly used methods to measure functional outcome in sarcoma patients [4]. However, due to the self-reported nature of their completion, they may be less reliable and vulnerable to bias. In the present study, the association between functional outcome measured by the MSTS and the TESS and objective measurements, such as the isokinetic dynamometer and 10-MWT, revealed a statistically significant association, which may support the usability of these questionnaires in sarcoma patients. The 9-HPT was found to be significantly associated with the MSTS, albeit no significant association was found between the 9-HPT and the TESS. The different results found between the MSTS and the TESS when compared to strength and the 9-HPT could be due to the fact that the MSTS is regarded as a measure of impairment of function, while the TESS measures functional disability [13,22]. No significant association was found between the MSTS and strength; however, a statistically significant association was found between strength and the TESS. The 6-MWT was not associated with either MSTS or TESS. It is, however, noteworthy that this study included both upper- and lower-extremity sarcoma patients, which could produce a bias since the walking ability is more impaired in lower-extremity sarcoma patients than upper-extremity sarcoma patients. Therefore, it is important to be aware of functional outcomes’ broader definition, and hence distinguish functional outcome in subcategories such as functional impairment or disability.

We used validated and reliable scales to measure different aspects of patients’ well-being including quality of life, insomnia severity, symptoms of depression and anxiety, as well as fatigue. Soft tissue sarcoma patients appeared to have a stable level of well-being throughout the first year. This is in contrast to other studies by Davidge et al. [18] and Davidson et al. [17], which found an improvement in the quality of life using European Quality of life – 5 Dimensions (EQ-5D). 

Soft tissue sarcoma patients were found to have increased anxiety scores as measured by the HADS prior to surgery and at one month after surgery, which seemed to subside throughout the first year following surgery. However, before and throughout the first year, most of the soft tissue sarcoma patients had anxiety scores within the normal range as defined by Snaith et al. [23]. Almost none had indications of depression before or after surgery. Similar results regarding anxiety and depression have previously been reported by Paredes et al. [24,25] and Refaat et al. [26].

Increased levels of sleep difficulties were observed both before and at one month after surgery in the soft tissue sarcoma population. Savard et al. [27] have previously determined that a cutoff of 8 on the ISI indicates sleep difficulties. A previous study by Refaat et al. reported periodic sleep difficulties in 24% of the sarcoma patients who had undergone limb-sparing surgery [26]. In another cancer population, Cha et al. [28] reported an association between sleep difficulties and patients’ overall quality of life, which underlines the importance of including a measure of sleep in research within cancer populations.

Fatigue has only rarely been reported in the literature regarding general soft tissue sarcoma population. Davis et al. reported that 33% of sarcoma patients (bone and soft tissue sarcoma, *N* = 80) experienced fatigue [15]. Patients generally evidenced low-to-moderate levels of fatigue across the different subscales of the MFI-20. Physical fatigue and reduced activity were the only scores to significantly change over time. This could possibly be related to the observed reduction in patients’ strength. Furthermore, lower-extremity sarcoma patients’ gait measured by the 10-MWT also seemed to negatively influence scores on general and physical fatigue and the reduced activity subscales. Upper-extremity sarcoma patients’ dexterity was significantly associated with anxiety measured by the HADS. Finally, soft tissue sarcoma patients’ exercise capacity was associated with symptoms of depression (HADS), general fatigue, physical fatigue, and reduced motivation. These findings confirm the important association between functional outcome and quality of life, including insomnia severity, fatigue, and psychological distress, which has been reported in previous studies [11,12,13,29].

Based on the findings of the present study, we recommend that future research investigates possible interventions to improve functional outcome in soft tissue sarcoma patients. This includes preventive interventions before surgery and rehabilitation after surgery in order to improve soft tissue sarcoma patients’ physical and mental health. Furthermore, knowledge about the physical and psychological effects of sarcoma and the treatment thereof provides valuable insights for clinicians and may be used to inform patients about possible short- and long-term effects following treatment.

A major strength of our study is the prospective study design with both pre- and post-surgery data available. However, there are also limitations. First, a potential bias is the number of participants who declined to participate (11 out of 40 patients), and out of these 73% simply stated that they did not wish to participate. Since it was not recorded why patients declined to participate it is unknown whether it was due to lack of physical or mental energy or because they did not anticipate any beneficial effect on their functional outcome. Second, missing data occurred due to the fact that some patients did not wish to participate at a given point in time, while others did not attend without any given reason. The multivariate analyses take missing data into account; however, when interpreting the univariate analysis, it is important to take missing data into account since these analyses only evaluate the available data.

## 4. Materials and Methods 

In this prospective observational study, all soft tissue sarcoma patients treated with first-time limb-sparing surgery at the Sarcoma Center at Aarhus University Hospital were included. Patients were recruited consecutively from November 2015 to May 2017. 

Patients were assessed with several functional outcome and self-reported measures at baseline prior to surgery, as well as 1 month, 3 months, and 1 year after surgery. Patients also responded to self-reported measures at two months post-surgery. The study was conducted in accordance with the Helsinki Declaration and approved by the local ethics committee (file no. 1−10−72−149−15) and approved by the Danish Data Protection Agency (file no. 1−16−02−650−15).

### 4.1. Study Population 

Inclusion criteria were patients aged 18 or above who were scheduled to undergo first-time limb-sparing surgery for newly diagnosed soft tissue sarcomas in the extremities; mentally capable for participation, and able to read and speak Danish. Patient exclusion criteria were (1) previous arthroplasty in a unilaterally affected extremity; (2) poor general condition defined as presence of severe cardiopulmonary diseases (DI519 and DJ894); (3) recurrence of the disease during the study period in, which case patients were excluded from the time of recurrence.

### 4.2. Outcome Measures

Figure 1 displays the study time points. At each scheduled appointment, objective measurements were undertaken. Functional outcome was evaluated with two questionnaires and three objective measurements.

The Danish version of the Musculoskeletal Tumor Society Score (MSTS) is a physician-completed, validated six-item, sarcoma-specific questionnaire in which three items are generic and three items are specific for either upper or lower extremities [5,30]. The MSTS was not conducted before surgery since it is not applicable preoperatively.

The Danish version of the Toronto Extremity Salvage Score (TESS) is a validated, patient-reported measure for assessing physical function in musculoskeletal tumor patients and exists in an upper- and lower-extremity version, which have 29 items and 30 items, respectively [6,31].

An isokinetic dynamometer provides a reliable and valid measurement of strength in patients by providing a constant velocity with accommodating resistance throughout the range of motion of the joint. [32,33]. Testing protocols were applied as described by Harbo et al. [10]. Normative strength data have been collected using this protocol in a previous study [10]. 

The 6-min walking test (6-MWT) is an easily performed test for measuring exercise capacity in patients and has been found to have satisfactory reliability and validity for use in cancer populations [34,35,36], furthermore, normative data have previously been established [37]. A 20–30 m hallway is required to perform the test. The patients are instructed to walk at their fastest pace covering as much distance as possible in the 6 min, during which the test supervisor gives standardized encouragements every minute [38].

The 10-m walk test (10-MWT) is a standardized and reliable method for assessing the maximum gait speed of a patient, including cancer patients [36,39,40]. The patients are instructed to walk at their maximum speed for 10 m. The middle six meters are timed, which allows the patients a two-meter acceleration/deceleration [39]. The 10-MWT was only used in patients with lower-extremity tumors.

The 9-hole peg test is a reliable measurement for determining the dexterity in the hand [41,42]. It has previously been used in cancer populations [43]. The 9-HPT consists of a plastic console with a shallow round dish for the pegs and nine holes on the other side of the console. The patient is asked to pick up the pegs one at a time using one hand only and put the pegs into the holes on the board in any order until all the holes are filled. Finally, the patient removes the pegs one at a time and returns them to the shallow dish on the board [41]. The score for the test is the time it takes the patients to complete these tasks.

Quality of life, insomnia severity, fatigue, and psychological distress was assessed with validated scales. Quality of life was assessed with the WHO-5 well-being index, which is a self-reported measure of well-being and has high validity for identifying well-being in patients [44]. The WHO-5 was developed by the WHO (World Health Organization) and consists of five items. Each item can be rated from 0 to 5, resulting in scores ranging from 0 to 25. It is recommended to convert this score to a percentage [44].

The Hospital Anxiety and Depression Score (HADS) is a reliable and valid self-reported questionnaire for assessing psychological distress (anxiety and depression) in cancer patients [45,46]. The HADS consists of 14 items which concern the patient’s experiences during the previous week. Each item is rated between 0 and 3. The HADS questionnaire contains two sub-scales (anxiety and depression), each based on 7 items, according to which each subscale ranges in score from 0 to 21. The lower the score, the less likely is the presence of anxiety or depression [45].

The Insomnia Severity Index (ISI) is a reliable and valid seven-item, self-reported questionnaire for the assessment of insomnia severity within the past 14 days in cancer patients. The patients rate their difficulties in falling asleep, maintaining sleep, waking up earlier than planned as well as their dissatisfaction of their current sleep pattern and how this affects their daily function. Each item is rated on a five-point scale (0–4) and reported as a score ranging from 0 to 28 [27,47]. 

The Multidimensional Fatigue Inventory (MFI-20) is a reliable and validated 20-item, patient-reported questionnaire developed for measuring fatigue in cancer patients. The MFI-20 consists of five subscales (four items per sub-scale): (1) general fatigue, (2) physical fatigue, (3) reduced motivation, (4) reduced activity, and (5) mental fatigue. Each item of the MFI-20 is rated on a five-point scale (1–5). Each subscale scoring ranges between 4 and 20, and a higher score indicates a higher level of fatigue [48,49].

The questionnaires were completed before the objective tests, and the objective tests were performed in the following order: (1) isokinetic dynamometer, (2) 10-MWT (lower-extremity patients) or 9-HPT (upper-extremity patients), and (3) 6-MWT.

### 4.3. Statistical Analyses 

Data were analyzed in Stata, version 15.0. Descriptive statistics were used for the patients’ clinical demographic. All variables were examined in order to determine the data distribution. *p*-values below 0.05 were considered statistically significant.

The isokinetic dynamometer and 6-MWT data were converted into a percentage of the expected outcome compared to the general population, using available equations for each measurement [10,37]. The average of the expected strength on both sides of the affected limb was calculated. 

Data from the 6-MWT, MSTS, TESS, WHO-5, HADS, ISI, and the MFI-20 did not have a Gaussian distribution. The median and interquartile range (IQR) were calculated for each of these variables. The Student’s *t*-test and the Wilcoxon rank sum test were used for evaluation of significant differences in outcomes before and after surgery. Furthermore, the Student’s *t*-test was used for analyzing differences in strength on the healthy and the disease-affected side. Repeated-measurement ANOVA was used as a multivariate analysis for significant change over time in outcomes, and the linear mixed model was used to investigate significant differences in strength and completion time of the 9-HPT between the disease-affected side and the healthy side. The results from these analyses were reported with F-statistic (F(df_time_, df_error_) = F-value, *p* = *p*-value). Associations between self-reported outcomes and objective measurements were evaluated using linear regression. When investigating the correlation between 9-HPTs and the self-reported outcomes, the average of the dominant and non-dominant sides was used. Moreover, an average of strength measured on both sides of the extremity was used to investigate the isokinetic dynamometer correlation to the different questionnaires.

## 5. Conclusions

Soft tissue sarcoma patients who have undergone limb-sparing surgery experienced a significant reduction in functional outcome in the disease-affected extremity during the first year after surgery. Furthermore, compared to the general population, their functional outcome was reduced. However, they evidenced improvement in their overall functional outcome during the first year following surgery.

The MSTS and the TESS were each associated with different aspects of functional outcome measured objectively and may therefore be used to evaluate various aspects of the functional outcome in soft tissue sarcoma patients. Future research would benefit from using these assessment methods in order to corroborate the present findings and to further investigate the potential physical long-term effects following treatment of sarcoma cancer. 

In general, soft tissue sarcoma patients reported minimal distress during the first year after surgery. However, distress and functional outcome were associated.

## Figures and Tables

**Figure 1 cancers-12-00463-f001:**
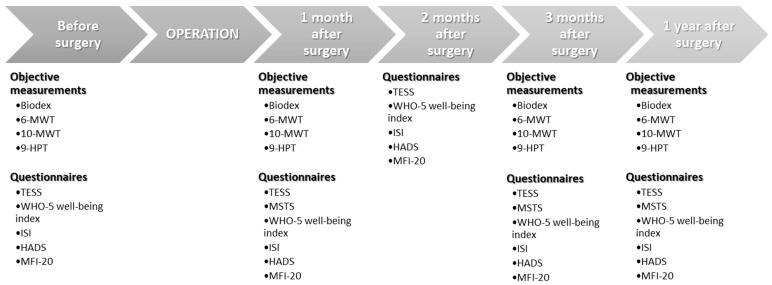
Time points for each measurement in the study.

**Table 1 cancers-12-00463-t001:** Baseline characteristics of the patients.

Characteristics	Patients (*n* = 29)	Characteristics	Patients (*n* = 29)
Age (in years + SD)	63 ± 17.1	Tumor grade (*n*, %)	
Range in age	25–86	Grade 1	4 (14%)
Gender (*n*, %)		Grade 2	11 (38%)
Male	15 (52%)	Grade 3	14 (48%)
Female	14 (48%)	Tumor margin (*n*, %)	
Location (*n*, %)		Wide	24 (83%)
Upper extremity	8 (28%)	Marginal	5 (17%)
Lower extremity	21 (72%)	Adjuvant treatment (*n*, %)	
Tumor size (in cm + SD)	8.85 ± 4.81	None	17 (59%)
Range in size	1–18	Radiotherapy (RT)	11 (38%)
Tumor depth (*n*, %)		RT + chemotherapy	1 (3%)
Subcutaneous	13 (45%)		
Subfascial	16 (55%)		

**Table 2 cancers-12-00463-t002:** The number of patients completing the objective measurements and the questionnaires at each time point.

Method	Before Surgery	1 Month after Surgery	2 Months after Surgery	3 Months after Surgery	1 Year after Surgery
	No. (%)	No. (%)	No. (%)	No. (%)	No. (%)
Objective measurements	28/29 (97%)	15/29 (52%)	N/A	19/27 (66%)	13/24 (45%)
Questionnaires	28/29 (97%)	20/29 (69%)	25/29 (86%)	24/27 (83%)	16/24 (55%)

N/A = not applicable.

**Table 3 cancers-12-00463-t003:** Functional outcome and quality of life measured by questionnaires.

Method	Score Range	Before Surgery (Baseline)		1 Month after Surgery (MSTS Baseline)			2 Months after Surgery			3 Months after Surgery			1 Year after Surgery		
		Median	IQR	Median	IQR	*p*	Median	IQR	*p*	Median	IQR	*p*	Median	IQR	*p*
TESS	0–100	96.5	88–100	78.5	57–93.5	<0.01 *	88	77–94	0.01 *	94	83.5–100	0.26	88	76–97.5	0.08
MSTS	0–100	N/A	N/A	76.5	51.5–95	N/A	N/A	N/A	N/A	97	85–100	0.03 *	93	80–100	0.11
WHO-5 well-being	0–100	70	42–92	68	54–74	0.36	76	64–84	0.63	74	56–96	0.47	80	68–92	0.24
HADS															
Anxiety score	0–21	5	3–7.5	4	1–5	0.12	2	0–5	0.03*	2	0.5–5	0.03*	2	1–5	0.03 *
Depression score	0–21	1	0–3	1	0–3	0.72	1	0–2	0.44	1	0–2	0.50	1	0–1	0.52
ISI	0–28	7	3–13	7	4–9	0.80	3	2–9	0.21	6	1–10	0.46	3.5	0.5–10	0.19
MFI-20															
General fatigue score	4–20	9	5.5–12.5	10.5	8–12	0.31	8	6–12	0.84	7	4.5–14.5	0.94	10	4.5–11.5	0.96
Physical fatigue score	4–20	8	6.5–13	13	11–18	<0.01 *	13	10–14	0.01 *	10	7.5–14.5	0.36	10	7–14.5	0.35
Mental fatigue score	4–20	8	4.5–11.5	7	5–9.5	0.63	6	4–9	0.22	6	4–8.5	0.33	6.5	4–9	0.28
Reduced activity score	4–20	9	4.5–13	11	9.5–13.5	0.07	10	8–13	0.30	9	6–12	0.82	9	5.5–10.5	0.73
Reduced motivation score	4–20	6.5	4–11.5	7	4.5–10	0.83	7	5–10	0.59	6	4.5–10.5	0.78	5	4–9	0.52

IQR: Interquartile range; TESS: Toronto Extremity Salvage Score; MSTS: Musculoskeletal Tumor Society Score; WHO-5: The World Health Organization-Five Well-Being Index; HADS: Hospital Anxiety and Depression Score; ISI: Insomnia Severity Index; MFI-20: Multidimensional Fatigue Inventory; * statistical significance (*p* < 0.05), *p*-value when compared to baseline by Wilcoxon rank sum test.

**Table 4 cancers-12-00463-t004:** Functional outcome measured by isokinetic dynamometer, the 10-m walk test (10-MWT), and the 9-hole peg test (9-HPT).

Method	Before Surgery			1 Month after Surgery			3 Months after Surgery			1 Year after Surgery		
	Mean	95% CI	*p*	Mean	95% CI	*p*	Mean	95% CI	*p*	Mean	95% CI	*p*
**Isokinetic dynamometer ^1^**												
Healthy side	80.27	73.46–87.09	0.60	83.06	75.16–90.96	<0.01 *	81.59	71.96–91.21	0.04 *	85.04	73.14–96.95	0.05 *
Disease-affected side	79.17	71.80–86.53		69.44	59.02–79.87		75.05	66.86–83.25		78.56	68.80–88.33	
Total average strength	79.72	72.96–86.48		76.25	67.78–84.72		78.32	69.94–86.70		81.80	71.42–92.19	
10-MWT (m/s)	2.12	1.91–2.34	Baseline	1.68	1.29–2.07	0.02 *	1.95	1.75–2.16	0.24	2.08	1.66–2.49	0.80
9-HPT (seconds)												
Healthy side	20.85	16.98–24.73	0.08	22.46	17.03–27.89	0.74	21.37	15.96–26.78	0.81	21.70	12.97–30.43	0.86
Disease-affected side	22.06	18.88–25.25		21.91	19.12–24.70		21.05	18.42–23.68		21.55	9.52–33.57	

*p*-value represents significance between: (1) the disease-affected and healthy side in the isokinetic dynamometer and the 9-HPT and (2) each point in time in the 10-MWT. ^1^ Isokinetic dynamometer outcomes are reported as a percentage of the expected strength when compared to normative data. * statistical significance (*p* < 0.05).

**Table 5 cancers-12-00463-t005:** Exercise capacity measured by the 6-min walking test (6-MWT).

Method	Before Surgery(Baseline)		1 Month after Surgery			3 Months after Surgery			1 Year after Surgery		
	Median	IQR	Median	IQR	*p*	Median	IQR	*p*	Median	IQR	*p*
**6-MWT:**	88.70	80.53–94.21	81.25	56.25–92.52	0.20	84.31	76.41–92.25	0.26	91.70	82.76–98.26	0.43

*p*-value when compared to baseline by Wilcoxon rank sum test.

**Table 6 cancers-12-00463-t006:** Linear regression analysis of objective measures’ correlation to questionnaires, where questionnaires’ outcome is a dependent variable and objective tests are an independent variable.

Method	Isokinetic Dynamometer		10-MWT (LE)		9-HPT (UE)		6-MWT	
	β	*p*	β	*p*	β	*p*	β	*p*
TESS	0.24	<0.01 *	13.09	<0.01 *	0.68	0.39	0.05	0.44
MSTS	0.24	0.17	28.80	<0.01 *	1.11	0.03 *	0.14	0.22
WHO-5 well-being	0.31	0.07	6.58	0.34	2.39	0.20	0.17	0.16
HADS								
Anxiety score	−0.01	0.89	0.75	0.43	−0.60	0.02 *	0.01	0.80
Depression score	−0.04	0.06	−0.08	0.91	0.03	0.91	−0.03	0.02 *
ISI	−0.04	0.35	−0.03	0.99	−0.73	0.08	0.01	0.72
MFI-20								
General fatigue score	−0.07	0.01 *	−2.16	0.05 *	−0.54	0.09	−0.05	0.01 *
Physical fatigue score	−0.13	<0.01 *	−3.00	0.01 *	0.01	0.99	−0.07	<0.01 *
Mental fatigue score	−0.03	0.18	0.51	0.55	−0.37	0.19	−0.02	0.28
Reduced activity score	−0.10	<0.01 *	−2.72	0.02 *	−0.32	0.27	−0.04	0.06
Reduced motivation score	−0.08	<0.01 *	−0.83	0.36	−0.32	0.23	−0.04	0.01 *

* Statistical significance (*p* < 0.05), LE = lower extremity patients, UE = upper extremity patients.

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
