# Peer review of "Changes in Functional Outcome and Quality of Life in Soft Tissue Sarcoma Patients within the First Year after Surgery: A Prospective Observational Study"

_cancers, 2020, doi:10.3390/cancers12020463_

Round 1

Reviewer 1 Report

In the manuscript by Saebye et al., titled, "Changes in functional outcome and quality of life in soft tissue sarcoma patients within the first year after surgery: a prospective observational study," the authors present their data from a prospective analysis of functional outcomes and quality of life for patients who have undergone limb-sparing surgery to remove a soft tissue sarcoma. 

The design of assessments was comprehensive and allowed for the connection between functional outcome and quality of life questionnaire assessments. One of the major criticisms is the bias of using self-reporting questionnaires and the bias that could be introduced by following a small cohort that had ~50% attrition by the end of the study. However, I felt like the authors have already addressed these criticisms in their discussion section, specifically the last paragraph. 

The one thing that the paper is missing is a simple take-away message about how this study can help patients who have had limb-sparing surgery to remove a soft tissue sarcoma. How does this knowledge make a difference for the patients? Especially because this paper is so clinically centric, it lacks the application of how this changes care for these patients in a clear and concise manner. 

Reviewer 2 Report

This is a prospective study of functional outcomes and quality of life as assessed prior to and after limb-sparing surgery in soft tissue sarcoma patients. The study is unique in the breadth of questionnaires and functional studies used in this study population and is able to highlight important, practice-changing outcomes. Specifically, it found that patients with soft tissue sarcomas are deconditioned even prior to surgery and may deserve increased physical therapy attention from diagnosis. It also found that, by one years' time, patients have mostly recovered from self-identified psychological stressors presumably associated with surgery.

Introduction: Could the authors provide a more robust explanation of the utility of MSTS, TESS, 10-MWT and other tests/questionnaires described. This could be done as a Table describing these tests and their application in this patient population. Results: The notation "F(4, 80)=9.02, p<0.01" will likely not be familiar to many readers. Please explain. The analysis of subjective versus objective measures should be clarified. Results section is slightly confusing to follow- please edit to emphasize meaningful comparisons.  Discussion: Well written and includes an elaborate limitations section. Could the authors elaborate on which tests they anticipate will provide benefit with future studies? Methods: may benefit from earlier introduction, especially if the readers are not sure of the different tests used and may benefit from learning about them earlier. Alternatively, a table of those tests, as suggested earlier, could have the same function. Conclusions: Edit to move this section after the discussion. Edit to include broad conclusions and future direction.

Round 2

Reviewer 2 Report

Edited manuscript has been reviewed. We appreciate the authors thoughtful responses.